# YOLO-T: Multitarget Intelligent Recognition Method for X-ray Images Based on the YOLO and Transformer Models

Mingxun Wang [1], Baolu Yang [1], Xin Wang [1,*], Cheng Yang [1], Jie Xu [1], Baozhong Mu [1], Kai Xiong [2] and Yanyi Li [3]

1   MOE Key Laboratory of Advanced Micro-Structured Materials, School of Physics Science and Engineering, Tongji University, 1239 Siping Road, Shanghai 200092, China
2   School of Information and Electrical Engineering, Zhejiang University City College, 51 Huzhou Street, Hangzhou 310015, China
3   College of Surveying and Geo-Informatics, Tongji University, 1239 Siping Road, Shanghai 200092, China
*   Correspondence: wangx@tongji.edu.cn

**Abstract:** X-ray security inspection processes have a low degree of automation, long detection times, and are subject to misjudgment due to occlusion. To address these problems, this paper proposes a multi-objective intelligent recognition method for X-ray images based on the YOLO deep learning network and an optimized transformer structure (YOLO-T). We also construct the GDXray-Expanded X-ray detection dataset, which contains multiple types of dangerous goods. Using this dataset, we evaluated several versions of the YOLO deep learning network model and compared the results to those of the proposed YOLO-T model. The proposed YOLO-T method demonstrated higher accuracy for multitarget and hidden-target detection tasks. On the GDXray-Expanded dataset, the maximum mAP of the proposed YOLO-T model was 97.73%, which is 7.66%, 16.47%, and 7.11% higher than that obtained by the YOLO v2, YOLO v3, and YOLO v4 models, respectively. Thus, we believe that the proposed YOLO-T network has good application prospects in X-ray security inspection technologies. In all kinds of security detection scenarios using X-ray security detectors, the model proposed in this paper can quickly and accurately identify dangerous goods, which has broad application value.

**Keywords:** X-ray; object detection; occlusion; YOLO; transformer; deep learning

## 1. Introduction

X-ray technology has been widely used in security inspection processes, which are essential to ensure public safety [1–3]. In these processes, perspective images can be obtained by scanning luggage or other containers using intense X-ray penetration, and it is possible to determine whether there is contraband inside the container without unpacking the contents. This method improves the speed of security inspection dramatically and is widely used in various locations, e.g., airports, bus stations, and high-speed railway stations [4,5]. X-ray security inspection image data have the following characteristics. (1) The irregularity of the shooting background, especially when the background is similar to the prohibited goods, will seriously interfere with object identification. (2) Objects in luggage are typically placed randomly, and overlapping occlusion increases recognition difficulty. (3) Different contraband come in diverse sizes, and the sizes of the same or similar types of contraband can also differ. Note that the image obtained by transmission scanning will be distorted, and the angle of placing the articles will change, which introduces significant difficulties in terms of effective object identification. Currently, most security inspection work relies heavily on human visual discrimination, and staff typically only have a few seconds to perform inspections. At peak operation times, staff can easily be affected by environmental noise, fatigue, and other factors, which lead to a high rate of missing prohibited goods and potential safety hazards. As deep learning technology has been used in the security detection field in recent years [6,7], the use of a deep learning target recognition network

can improve the automation of security inspection processes significantly and is of great significance to improve the inspection efficiency and recognition accuracy.

X-ray security inspection contraband detection is performed to locate and classify contraband in the image. Generally, traditional machine learning target detection methods use sliding windows to locate target objects [8,9], and classifiers are employed to identify the target objects. Unfortunately, the feature extraction ability of such methods is weak, it is difficult to capture detailed information about the target, and detection is poor for contraband objects that are blocked and placed at different angles. However, the network structure of the deep learning target detection algorithm is deep, and the location and classification accuracies can be improved by fusing different features between the network's layers. Currently, deep learning target detection algorithms can be divided into two-stage and one-stage detection algorithms. Two-stage algorithms are represented by the R-CNN method [10], which first selects several candidate regions using independent algorithms. Then, the features of the candidate regions are extracted through networks. Finally, this method locates and classifies feature vectors using support vector machines and regressors. (2) A one-stage algorithm transforms the target detection task into a regression problem and predicts the target using an end-to-end network. This technique is sufficiently fast to realize real-time detection. YOLO [11] and SSD [12] are representative examples of one-stage detection algorithms.

Deep learning technology has recently been used in X-ray security inspection processes. Akcay et al. first introduced deep learning technology to baggage classification of X-ray images, verified that deep learning is more suitable than other methods for X-ray image classification tasks, and proposed an anomaly detection model that uses a conditional generation countermeasure network to train a large number of X-ray images without contraband so that when the contraband appears, it will be marked as an abnormal condition in the image. On this basis, anomaly detection is performed on images with contraband, and good results have been achieved in several studies [13–16]. In addition, Gaus et al. [17] employed a two-stage detection method with a double CNN structure to find the region of interest in the image, to obtain the corresponding location information, and then to classify the anomaly. Here, the detection accuracy was improved slightly by combining target detection and anomaly detection techniques. However, the network structure of this method was complex, detection speed was slow, and it was difficult to distinguish objects with similar shapes. Contraband hidden in suitcases are generally small, e.g., guns and knives, and detecting small dangerous objects is challenging. In addition, an X-ray image differs from a color image, and the details and color information of this kind of data are lost. Thus, most deep learning network models do not recognize overlapping objects effectively. Thus, Liu et al. [18] proposed a front and back background segmentation method to remove most of the background containing useless information according to the brightness differences between the front and back scenes of the X-ray image. Hassan et al. [19] used the contour information of the object in the X-ray image, generated a series of tensors using cascade structure tensor technology, provided target suggestions, and then conducted target detection using a convolution neural network based on the contour information and candidate targets. Note that implementing an attention module can also effectively solve the occlusion problem. Li et al. [20] combined a semantic segmentation network with the mask R-CNN, where they took the semantic segmentation result as the soft attention mask of the mask R-CNN, which improved the detection accuracy of overlapping objects. In addition, Zhang et al. [21] proposed the XMC R-CNN model. They used the X-ray material classifier and organic and inorganic stripping algorithm for the first time to solve the problem of overlapping contraband in X-ray images. Although many previous studies have investigated the occlusion and overlap of targets in X-ray images, recognition accuracy and computation speed still limit the application of such technology in the security detection field. In summary, further research is required to develop deep learning networks that exhibit high recognition accuracy, good stability, and fast detection speeds.

Given that traditional machine learning algorithms cannot handle the increasingly complex security environment. Based on an existing deep learning target detection network under the YOLO framework, we propose an innovative YOLO-T network structure to detect dangerous small-scale goods under complex environmental conditions, e.g., multiple targets and occlusion. The proposed YOLO-T model provides a complete method for target identification of dangerous goods based on X-ray images in the security inspection field. Through our experiments, the detection accuracy of the YOLO-T network reached 97.73%, and the best results were obtained in the comparison experiments with various networks in YOLO series. In addition, the proposed model solves the problems of difficulty and low precision when detecting small target objects in the security inspection field and improves the detection efficiency. From the perspective of practical application, public transport and airports are equipped with X-ray security detectors. The use of the depth learning model can quickly and accurately identify dangerous goods from the images taken by the X-ray security detector. When the flow of people is large and the safety detection task is heavy, rapid and automatic detection of dangerous goods is of great significance.

This article's structure is as follows. The first part introduces the research status of X-ray image security detection. The second part introduces the basic principle of the YOLO-T method. In the third part, a comparative experiment is carried out between the YOLO-T method and other methods. The last part is the summary and prospect of the full research.

## 2. Principle and Method

### 2.1. Object Detection Based on YOLO Deep Learning Network

The primary principle of the YOLO technique is to divide an image into a $7 \times 7$ grid. If the center of an object falls in a grid, the grid is responsible for predicting the object. Here, each grid must predict two bounding boxes, and each bounding box predicts $(x, y, w, h)$ and the confidence level, where $(x, y)$ represents the position of the center point of the bounding box, and $(w, h)$ represents the width and height of the bounding box after normalization relative to the entire image. The confidence level includes the probability that the bounding box will contain the target $Pr(object)$ and the accuracy of the bounding box (the intersection ratio of the prediction box and the actual box) $IOU_{pred}^{truth}$, i.e., $confidence = Pr(object) \times IOU_{pred}^{truth}$. Note that a high confidence level indicates that there is a target in the bounding box and that the position is accurate. In contrast, a low confidence level indicates that there may be no target or, if there is a target, there is a large position deviation. In addition, each grid must predict the target categories, and the probability of each category is denoted $P(C_i|object)$. Initially, 20 categories can be predicted. Finally, the predicted object information is a $7 \times 7 \times 30$ tensor. The prediction frame and category information extracted from it are processed by non-maximum suppression to output the prediction result for the given image.

The network structure of YOLO is motivated by the GoogleNet model. The YOLO network comprises 24 convolution layers, four max-pooling layers, and two fully connected layers. The convolution layer of $3 \times 3$ is connected after the restoration layer of $1 \times 1$ to replace the GoogleNet's Inception module. The YOLO network structure is illustrated in Figure 1.

YOLO employs the mean square error as the loss function, which comprises the positioning loss, confidence loss, and classification loss. Here, the positioning loss is used to measure the bounding box's central coordinate error and the bounding box's width and height error. Note that smaller target bounding boxes are disturbed by deviations caused by positioning loss. To make the model pay focus to the bounding box containing the target, the confidence loss weight of the bounding box containing the target is typically set to 1, and the confidence loss weight of a bounding box without the target is 0.5. After setting the above parameters, the classification loss is calculated only when targets are present in the grid. Here, each grid has multiple bounding boxes but corresponds to only a single category; thus, only the box with the giant IOU of the ground truth is selected for training. The loss function is divided into the following three expressions.

Positioning loss formula:

$$\lambda_{coord}\sum_{i=0}^{S^2}\sum_{j=0}^{B}1_{ij}^{obj}\left[(x_i-\hat{x}_i)^2+(y_i-\hat{y}_i)^2\right]+\lambda_{coord}\sum_{i=0}^{S^2}\sum_{j=0}^{B}1_{ij}^{obj}\left[\left(\sqrt{w_i}-\sqrt{\hat{w}_i}\right)^2+\left(\sqrt{h_i}-\sqrt{\hat{h}_i}\right)^2\right] \tag{1}$$

Confidence loss formula:

$$\sum_{i=0}^{S^2}\sum_{j=0}^{B}1_{ij}^{obj}\left(C_i-\hat{C}_i\right)^2+\lambda_{noobj}\sum_{i=0}^{S^2}\sum_{j=0}^{B}1_{ij}^{noobj}\left(C_i-\hat{C}_i\right)^2 \tag{2}$$

Classification loss formula:

$$\sum_{i=0}^{S^2}1_i^{obj}\sum_{c\in class}\left(p_i(c)-\hat{p}_i(c)\right)^2 \tag{3}$$

Here, $S$ represents the grid, and $B$ represents the number of prediction boxes; 1 is an indicating function with only two values, i.e., 0 and 1. If a target is present in grid $i$, $1_i^{obj}$ is set to 1; otherwise, it is set to 0. $1_{ij}^{obj}$ indicates that the $j$th prediction box in grid $i$ predicts the grid. The $(x_i, y_i)$ coordinates represent the center of the box relative to the bounds of the grid cell, and $(\hat{x}_i, \hat{y}_i)$ represents the position coordinate of the corresponding actual box. In addition, $w_i, h_i, \hat{w}_i, \hat{h}_i$ represent the width and height of the prediction box and the actual box relative to the entire image. $C_i$ is the confidence score.

Based on the above principles, the YOLO v2 network comprises the DarkNet-19 feature extraction and detection network. YOLO v3 employs the DarkNet-53 CNN as the backbone network. To avoid the gradient vanishing problem caused by the deep structure of the network, the network draws lessons from the structure of the residual network and the feature pyramid network concept. The YOLO v4 network comprises a backbone, a neck, and a head structure, where CSPDarkNet-53 is employed as the backbone network. In other words, the YOLO v4 network implements cross-stage partial connections based on DarkNet-53. Currently, the abovementioned YOLO network models are widely used in target-detection tasks [22–24].

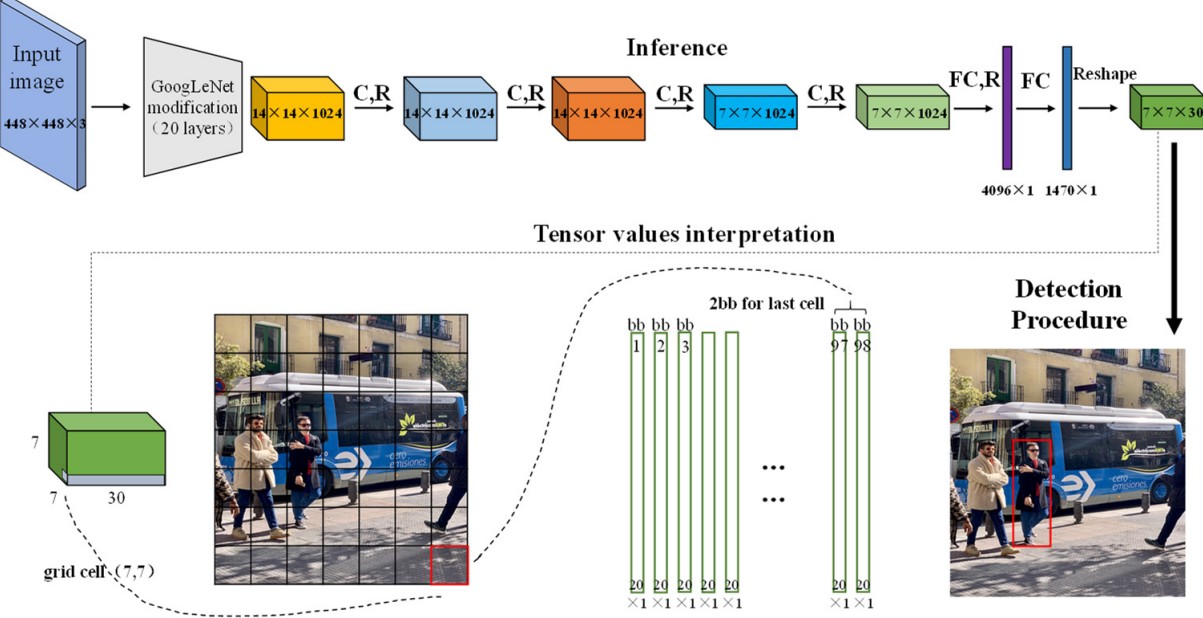

**Figure 1.** Schematic diagram of YOLO deep learning network structure.

### 2.2. Build YOLO-T Target Detection Network

The transformer structure was proposed by Google's deep learning team [25]. It is an encoder–decoder network structure based on an attention mechanism, as shown in Figure 2. The so-called coding transforms the input sequence into a fixed length vector, and the decoding process converts the previously generated fixed vector into an output sequence.

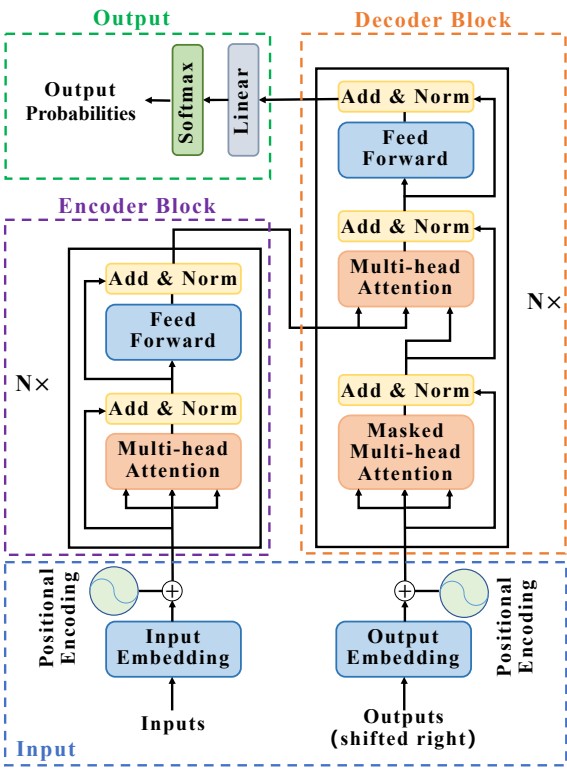

**Figure 2.** Transformer network structure diagram.

Sequence data (vector) are input, position encoding is performed, and then, the encoder is entered after splicing. The location coding processes are expressed as follows.

$$PE_{(pos,2i)} = \sin\left(\frac{pos}{10,000^{\frac{2i}{d_{model}}}}\right) \tag{4}$$

$$PE_{(pos,2i+1)} = \cos\left(\frac{pos}{10,000^{\frac{2i}{d_{model}}}}\right) \tag{5}$$

Here, $d_{model}$ represents the total dimension of the input vector, *pos* represents the position in the vector, and *i* represents the number of dimensions in the vector. The parameter 10,000 here is the specified parameter in the location coding formula. Specifically, it refers to the geometric series from $2\pi$ to $10,000 \times 2\pi$ formed by wavelength in the process of trigonometric function operation, which is related to the series expansion.

The encoder block shown in Figure 2 comprises six stacked encoders, and each encoder comprises a multi-head self-attention module and a fully connected feedforward neural network. Note that the attention model was first employed in the machine translation field and has played an important role in computer vision tasks in recent years. A schematic diagram of the attention concept is shown in Figure 3.

As shown in Figure 3, the elements in the source comprise a series of key and value data pairs. Given an element query in the target, the weight of the value corresponding to each key is obtained by calculating the similarity between the query and each key. The value is then weighted and summed to obtain the final attention value. Here, the greater

the weight, the more focus on its corresponding value. Using the attention mechanism, we can filter out essential information from a large amount of data while ignoring unimportant information. The calculation of the attention mechanism is expressed as follows.

$$Attention(Q, K, V) = \text{softmax}\left(\frac{QK^T}{\sqrt{d_k}}\right)V \tag{6}$$

Here, $d_k$ represents the dimension of $K$. Note that the self-attention mechanism only uses a single set of $Q, K, V$ to calculate the attention value, while the multi-head self-attention mechanism uses multiple sets of $Q, K, V$ to calculate the attention value. Finally, the resulting multiple matrices are spliced as follows.

$$MultiHead(Q, K, V) = \text{Concat}(head_1, \ldots, head_i,)W^o \tag{7}$$

$$head_i = Attention\left(QW_i^Q, KW_i^K, VW_i^V\right) \tag{8}$$

Here, $QW_i^Q, KW_i^K, VW_i^V$ is the mapping weight matrix of $Q, K, V$ in the $i$-th multi-head attention mechanism, and $W^o$ is the weight matrix.

The matrix X output by the multi-head attention module is input to the feedforward fully connected neural network after residual and normalization. The calculation formula is given as follows.

$$FFN(X) = \max(0, XW_1 + b_1)W_2 + b_2 \tag{9}$$

Through the feedforward neural network, matrix $X$ changes back to the dimension before the input encoder and enters the decoder. Six decoders are stacked on the decoder block. Each decoder has more masked multi-head attention than the encoder block. In practice, some values are masked to not affect the updating parameters. This is done to block the following information. For a sequence, at time t, the decoder output should only depend on the output before t, and the information after t must be hidden. After a linear transformation process, the output of the decoder obtains the probability distribution of the output through the SoftMax function, and the prediction with the most significant corresponding probability is selected as the final prediction result.

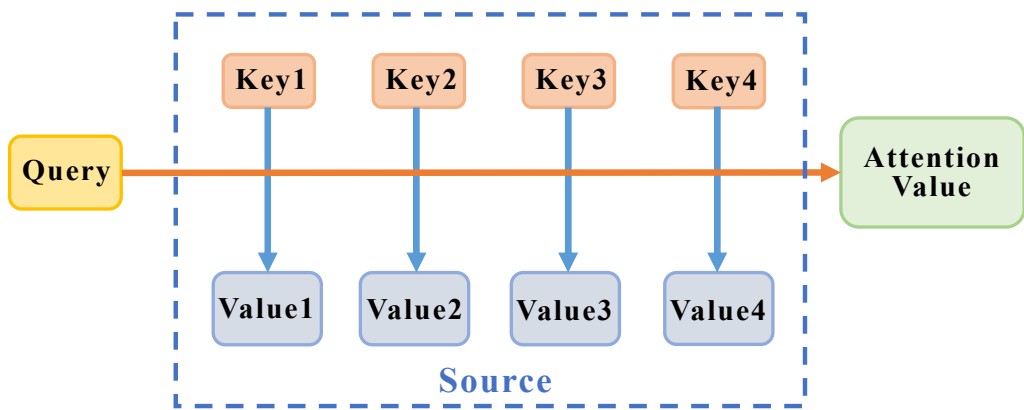

**Figure 3.** Schematic diagram of the attention concept.

The full name of the proposed YOLO-T network is YOLO-Transformer. As a deep learning network designed specifically for the X-ray security detection task, the main structure of the proposed YOLO-T network is shown in Figure 4. When designing the network, we replaced the DarkNet-53 network as the backbone in the YOLO v3 network with Swin transformer structure [26] while retaining the neck and head parts of YOLO v3.

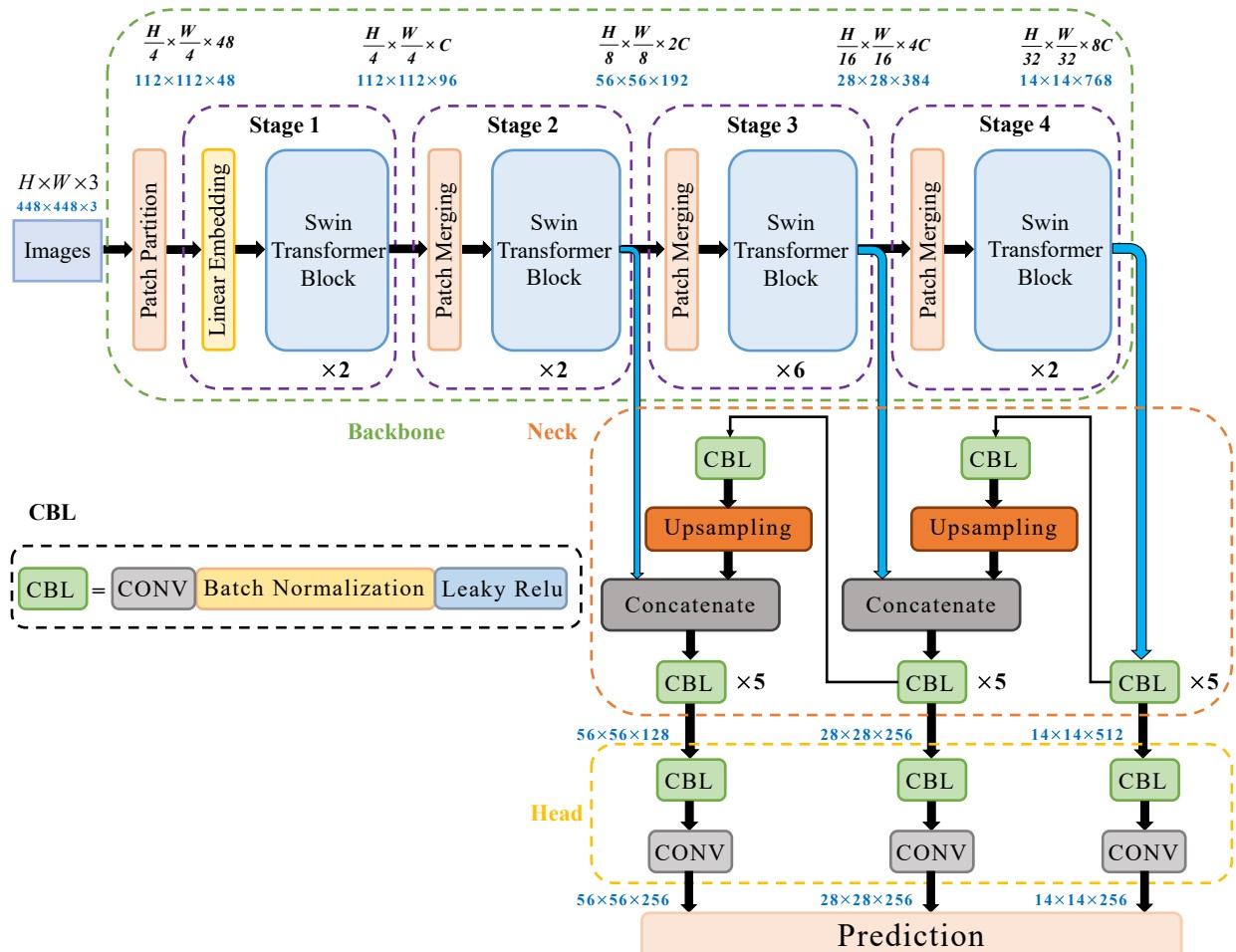

**Figure 4.** Schematic diagram of YOLO-Transformer network structure.

As shown in Figure 4, the network structure of the Swin transformer backbone is the key to the entire YOLO-T network. First, the input $H \times W \times 3$ image is divided into equal size nonoverlapping $N \times (P^2 \times 3)$ patches through the patch partition module. The patch of each $(P^2 \times 3)$ is recorded as a patch token, with a total of N patch tokens. Here, P is 4, and the flattened dimension of each patch is 48, $N = \frac{H}{4} \times \frac{W}{4}$. Then, linear embedding projects the $\frac{H}{4} \times \frac{W}{4} \times 48$ tensor to any dimension C to obtain a tensor of dimension $\frac{H}{4} \times \frac{W}{4} \times C$. It is then sent to several shifted window transformer blocks (STB) with improved self attention. The first STB and the linear embedding layer form stage 1. As the network depth increases, the number of operation tokens through the patch-merging layer decreases gradually. After the first patch is merged and spliced, the number of patch tokens is reduced to one-quarter of the original, i.e., $N = \frac{H}{8} \times \frac{W}{8}$, and the dimension becomes 4C. The dimension is reduced to 2C through a linear layer and then sent to an STB in stage 2 for feature conversion. After each stage, the number of channels is expanded twice, and then, the $\frac{H}{32} \times \frac{W}{32} \times 8C$ tensor is output. The above discussion focuses on the backbone part of this network. We take a $448 \times 448 \times 3$ data image input as an example to show the network structure. Then, the remaining neck and head parts of the network maintain the basic YOLO structure, as shown in Figure 4. It takes the RGB image as an example. For detecting single-channel X-ray images, the number of image channels is changed to one during training.

The STB structure in each stage is shown in Figure 5 [26]. It retains other parts of the classical transformer structure and replaces the standard multi-head self-attention (MSA) module with the multi-head self-attention modules W-MSA and SW-MSA based on the shift window. The STB comprises an MSA module based on the shift window. Each MSA module and each multilayer perceptron (MLP) uses the LayerNorm (LN) previous layer,

and all use the residual connection. The STBs with shift window division are calculated as follows.

$$\hat{z}^l = W - MSA\left(LN\left(z^{l-1}\right)\right) + z^{l-1} \tag{10}$$

$$z^l = MLP\left(LN\left(\hat{z}^l\right)\right) + \hat{z}^l \tag{11}$$

$$\hat{z}^{l+1} = SW - MSA\left(LN\left(z^l\right)\right) + z^l \tag{12}$$

$$z^{l+1} = MLP\left(LN\left(\hat{z}^{l+1}\right)\right) + \hat{z}^{l+1} \tag{13}$$

Here, $\hat{z}^l$ represents the (S)W-MSA module output characteristics of the *l*-th block, and $z^l$ represents the MLP module output characteristics of the *l*-th block.

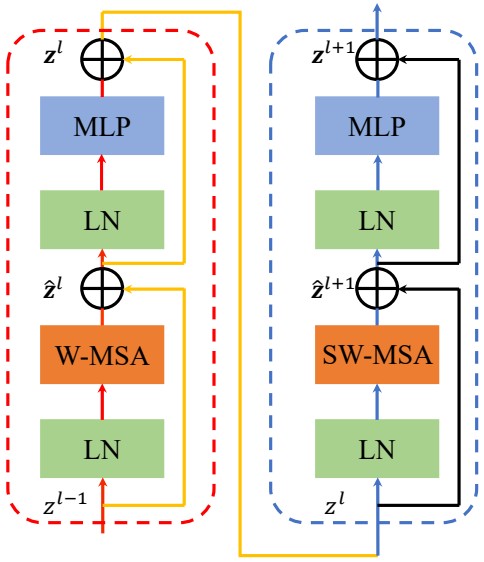

**Figure 5.** Two successive Swin transformer blocks.

Different from the existing structure, we couple this structure with YOLO structure and apply its migration to the target detection of X-ray images. This part of the work is mainly reflected in the actual program structure and preparation. After our subsequent experiments, we found that this structure can effectively suppress the false detection problem caused by overlapping and background interference in the detection process. In summary, to the best of our knowledge, the proposed YOLO-T network is the first to integrate the attention mechanism and transform structure for application in the X-ray security detection field. By designing several improved self-attention STB structures, the proposed YOLO-T network has higher resource utilization efficiency, greatly improved processing speed compared to traditional YOLO series networks, and it effectively solves the problem of X-ray image target detection under interference from various environmental conditions. In addition, the transformer structure can be divided into multiple blocks (similar to the CNN), and feature information of different resolutions can be extracted between blocks at different levels; thus, the proposed YOLO-T network exhibits high accuracy and fast detection speed in the multitarget detection task. In summary, the proposed YOLO-T network obtains better detection results when applied to X-ray image target detection than to conventional traditional YOLO networks.

## 3. X-ray Image Recognition Experiment

### 3.1. Experimental Dataset

Most deep learning object detection models are based on optical color images. Public optical image datasets are easy to obtain, and the production costs of such datasets are low;

however, it is challenging to obtain X-ray images. Current, X-ray datasets applicable to the detection of contraband include the GDXray and SIXray datasets [27,28].

The GDXray dataset was the first public large-scale dataset of X-ray images (8150 X-ray baggage images). The GDXray dataset contains images of guns, swords, and other blade contraband. The data in the GDXray dataset are grayscale images, in which the target contour is clear and easy to distinguish, the background is simple, and object overlap and occlusion are less. Example images from the GDXray dataset are shown in Figure 6a.

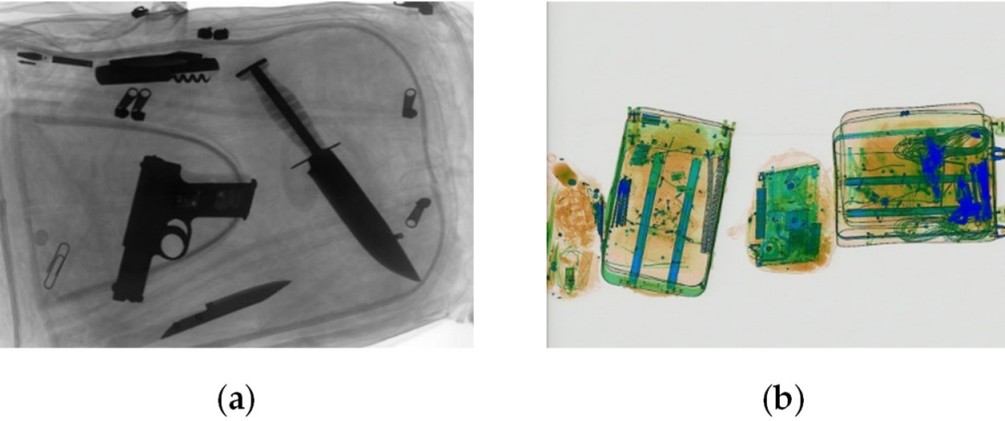

**(a)**                                                 **(b)**

**Figure 6.** Example images: (**a**) GDXray dataset; (**b**) SIXray dataset.

The SIXray dataset contains more than one million X-ray images, including six types of targets, i.e., guns, knives, wrenches, pliers, scissors, and hammers, and 8929 marked images containing these targets. The stacking randomness of the target objects is quite large in the SIXray dataset, the image backgrounds are complex, and there are instances of severe overlap and occlusion. Example images from the SIXray dataset are shown in Figure 6b.

In this study, we created the GDXray-Expanded dataset. This dataset is integrated according to the real-world dangerous goods encountered during the security inspections at stations and airports in mainland China, primarily including guns, scissors, knives, and four types of explosives of different shapes and sizes. It is also the primary dataset mainly used in our experiments. Example images from the GDXray-Expanded dataset are shown in Figure 7. X-ray transmission experiments obtained the GDXray-Expanded dataset images. There are few dangerous species in the GDXray dataset, the image background is simple, and there are almost no occlusion and overlap between objects, which is not conducive to detection in the actual security inspection scene; thus, we took a large number of images ourselves. Fundamentally, considering that the GDXray dataset is a real gray image dataset, it is completely consistent with the real situation of the security camera.

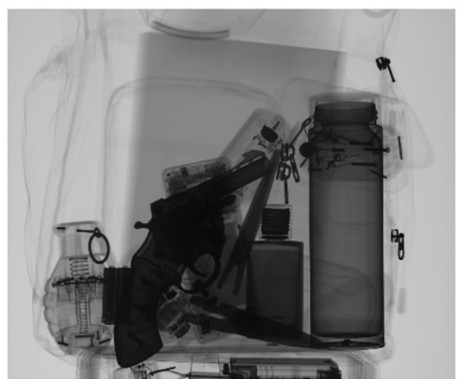 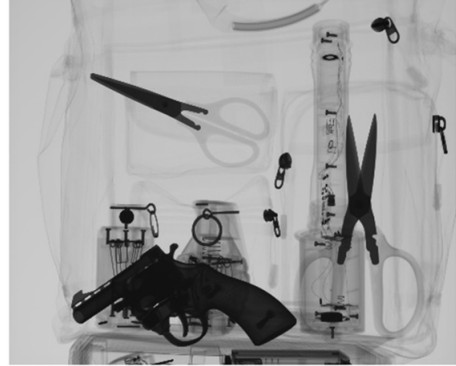

**Figure 7.** Example images from the GDXray-Expanded dataset.

Our experiment was based on the basic principles of X-ray transmission imaging. The experimental system comprised an X-ray source and a detector. Here, the X-ray source uses electrons to bombard a tungsten target to generate continuous and characteristic X-rays with an energy of 30–120 keV. More specifically, we use amorphous silicon detectors here. Amorphous silicon detectors are mainly composed of a scintillator, light sensing panel and charge readout circuit. When the X-ray is incident, the scintillator converts the attenuated X-ray passing through the object into visible light and then converts the visible light into an electrical signal through the amorphous silicon photodiode sensor array, forming a storage charge on the capacitance of the photodiode itself. After that, the control circuit scans and reads out the storage charge of each pixel, outputs digital signals after signal amplification and A/D conversion, and transmits them to the computer for image processing to form an X-ray image. To carry out our X-ray transmission experiment, it was necessary to place the target object in the front of the flat panel detector and set the exposure time of the radiation source to obtain a high-resolution two-dimensional grayscale image. Figure 8a,b show the set-up for penetration imaging and the experimental system, respectively. During this experiment, the X-ray source voltage was set to 120 kV, and the detector integration time was 1 s.

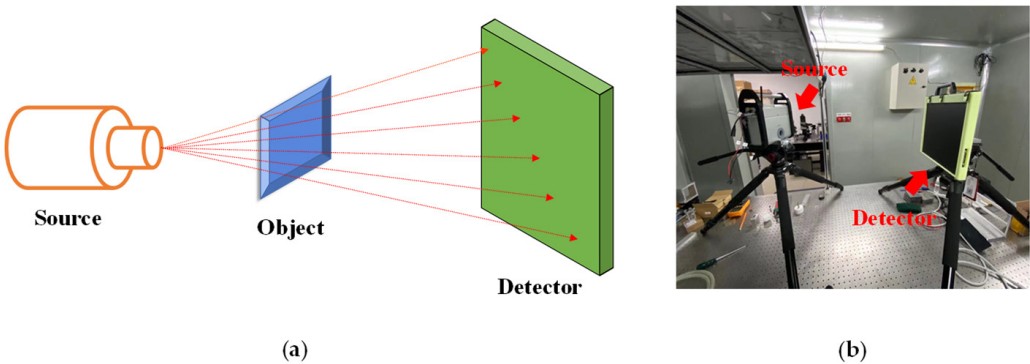

(**a**)                                           (**b**)

**Figure 8.** (**a**) Set-up for penetration imaging; (**b**) experimental system.

### 3.2. Data Preparation and Preprocessing

The target dangerous goods include gun, scissors, knives, and several types of explosives. According to the shape, explosives are divided into four categories, i.e., grenade-handle (GH), grenade-normal (GN), grenade-rectangle (GR), and grenade-tube (GT). The shapes of the seven dangerous goods are shown in Figure 9. We selected 1158 knife images, 1056 gun images, and 64 scissors images from the GDXray dataset. In addition, grayscale images of explosives were taken from the laboratory. Considering the limited number of images taken in the laboratory and the complexity of the background of the actual security inspection scene, we selected 138 background images that did not contain the target from the GDXray dataset and took many complex background images without the target to be tested. Here, we used image fusion technology to fuse a single dangerous good with the background image, and the resulting fused grayscale image was very similar to the real image. We also applied image enhancement techniques, e.g., vertical flip, horizontal flip, brightness adjustment, and Gaussian noise, to expand the dataset to 10,000 images. The dataset was divided into training and test sets at a ratio of 3:1. The training and test sets both contained the seven types of items to be tested. Note that the experimental GDXray-Expanded dataset was constructed to facilitate training of deep learning models.

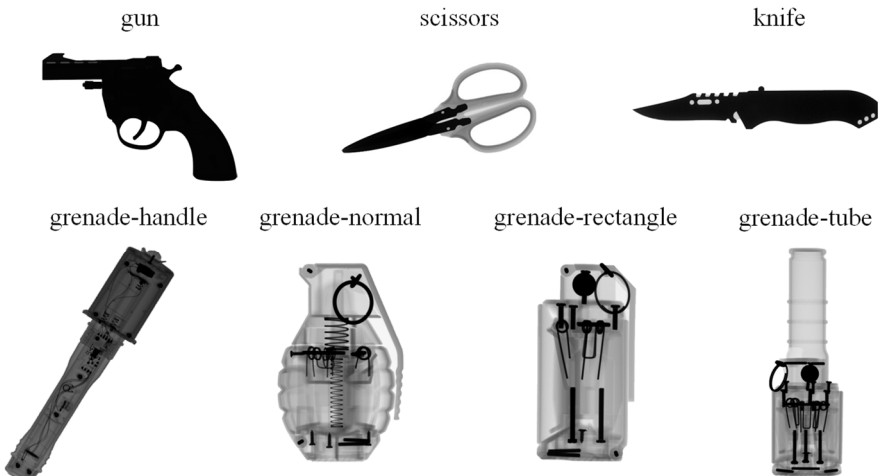

**Figure 9.** Examples of target dangerous goods.

### 3.3. Experimental Environment

The hardware environment used in this experiment is summarized as follows. The experiments were run on a personal computer with an Intel Core i7-11800 h CPU with 32 GB of memory and a 512 GB solid state disk. The graphics card was an NVIDIA GeForce RTX3090 with 24 GB of video memory. The software environment is summarized as follows. The operating system was Windows 10. Python 3.8 was used as the primary programming language, the Darknet and PyTorch deep learning frameworks were used to train the network, and the OpenCV library was used for visual processing. Darknet is an open source deep learning framework developed by Joseph Redmon, the author of YOLO. For information about Darknet, refer to the link: https://pjreddie.com/darknet/ (accessed on 19 October 2022)

The training parameters of the deep learning networks were set as follows. YOLO v2, YOLO v3, and YOLO v4 were trained and tested using the Darknet framework, the batch size was set to 16, momentum was set to 0.9, and the weight decay was set to 0.0005. Comparison was made by changing the learning rate (0.001, 0.003, 0.005) and the number of iterations (9000, 14,000). The proposed YOLO-T was trained and tested using the PyTorch framework. Here, the learning rate was set to 0.0001, the batch size was set to 16, and the number of epochs was set to 500. The optimizer used for training the model was Adam with decoupled weight decay (AdamW). It is an improved algorithm based on the regularization of Adam + L2. L2 regularization needs to add a regularization term to loss, to calculate the gradient, and then to back-propagate. AdamW directly adds the gradient of the regularization term to back-propagation, which improves the calculation efficiency.

### 3.4. Object Intelligent Identification Process Based on YOLO-T

The flow of the experimental methodology is illustrated in Figure 10. As can be seen, the experimental process includes three main parts, i.e., (1) dataset preprocessing and data enhancement, (2) training the YOLO network, and (3) testing the trained model. (1) In the dataset preprocessing and data enhancement process, the marked X-ray dangerous goods dataset was initially divided into training and testing sets at a ratio of 3:1. Then, the training data were enhanced using various techniques, e.g., mosaic enhancement, vertical and horizontal flipping, and brightness adjustment. (2) The YOLO network model was then trained. Here, the input image was adjusted to a specific resolution, and the adjusted image was input to the YOLO v2, YOLO v3, YOLO v4, and YOLO-T network models to train the dataset to obtain the dangerous goods detection model. (3) Then, we performed dangerous goods identification and evaluated each model on the test set.

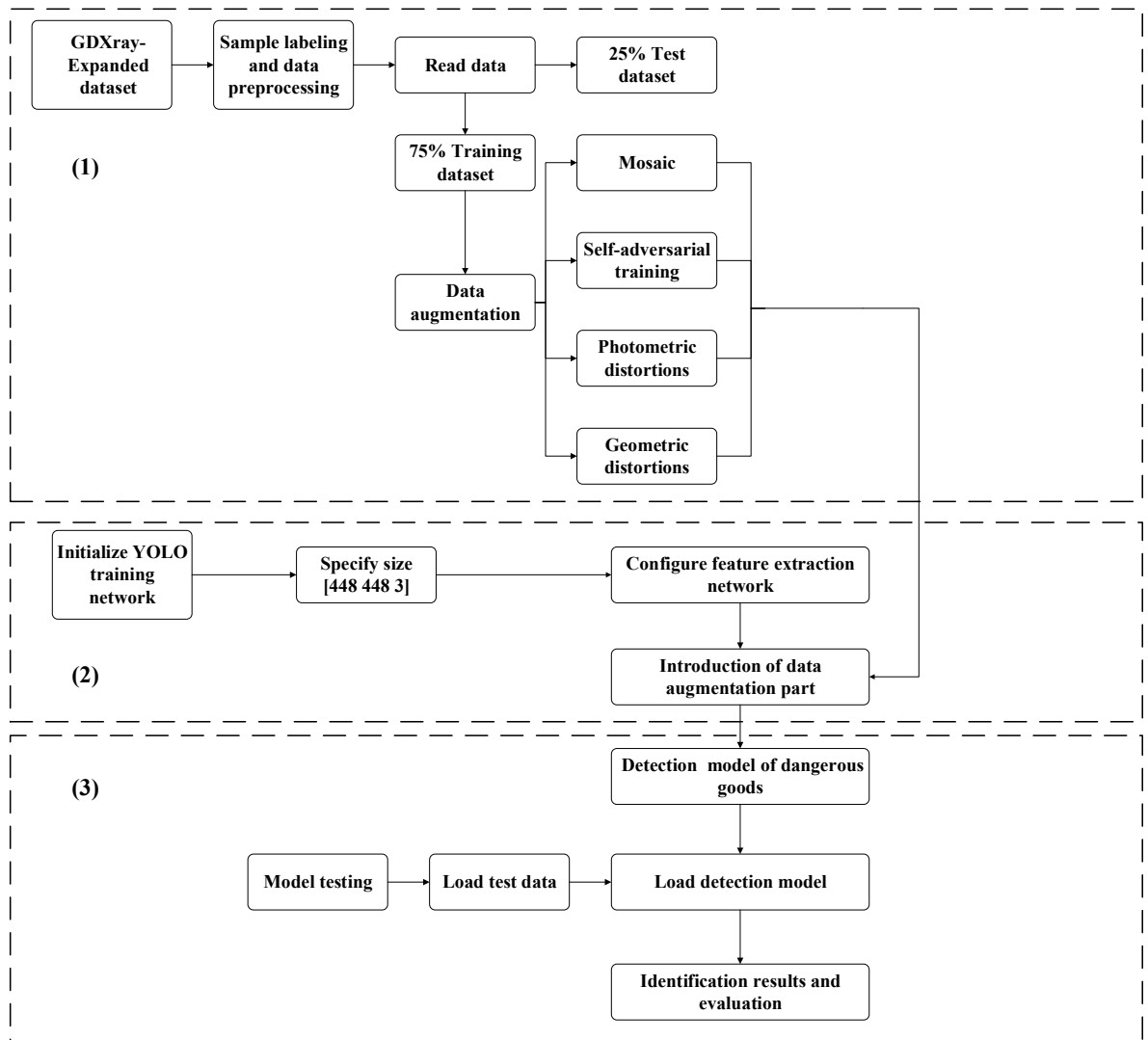

**Figure 10.** Experimental methodology: (1) dataset preprocessing and data enhancement; (2) training the YOLO network; (3) testing the trained model.

## 4. Experimental Results and Discussion

### 4.1. YOLO Recognition under Different Learning Rates and Iterations

We trained the YOLO v2, YOLO v3, and YOLO v4 network models on the GDXray-Expanded dataset using different learning rates and numbers of iterations. Here, 18 experimental models were used to evaluate the corresponding test set, and the detection accuracy and mAP of the seven kinds of dangerous goods were calculated and compared.

As shown in Figure 11, the detection accuracy of YOLO v2 for each type of dangerous goods under different learning rates and iterations was approximately 90% and relatively stable. However, the mAP value of YOLO v2 was slightly lower than that of YOLO v4 when the learning rate was set to 0.001, and it demonstrated the best performance in all cases in terms of mAP. Thus, the YOLO v2 network model exhibited the best overall performance. As shown in Figure 11c,d, when the learning rate was set to 0.003, the overall effect of YOLO v3 and YOLO v4 was general, especially since the detection accuracy for GH explosives and scissors was low. As shown in Figure 11e,f, when the learning rate was set to 0.005, the convergence effect of the YOLO v3 and YOLO v4 models was poor. The mAP of the three network models was 90.07% for YOLO v2 (Figure 11f), 81.26% for YOLO v3 (Figure 11d), and 90.62% for YOLO v4 (Figure 11a). However, in terms of parameter adjustment, we

observed significant fluctuation in the accuracy of the YOLO v4 network model. Thus, we determined that the YOLO v2 network model demonstrated the best overall performance.

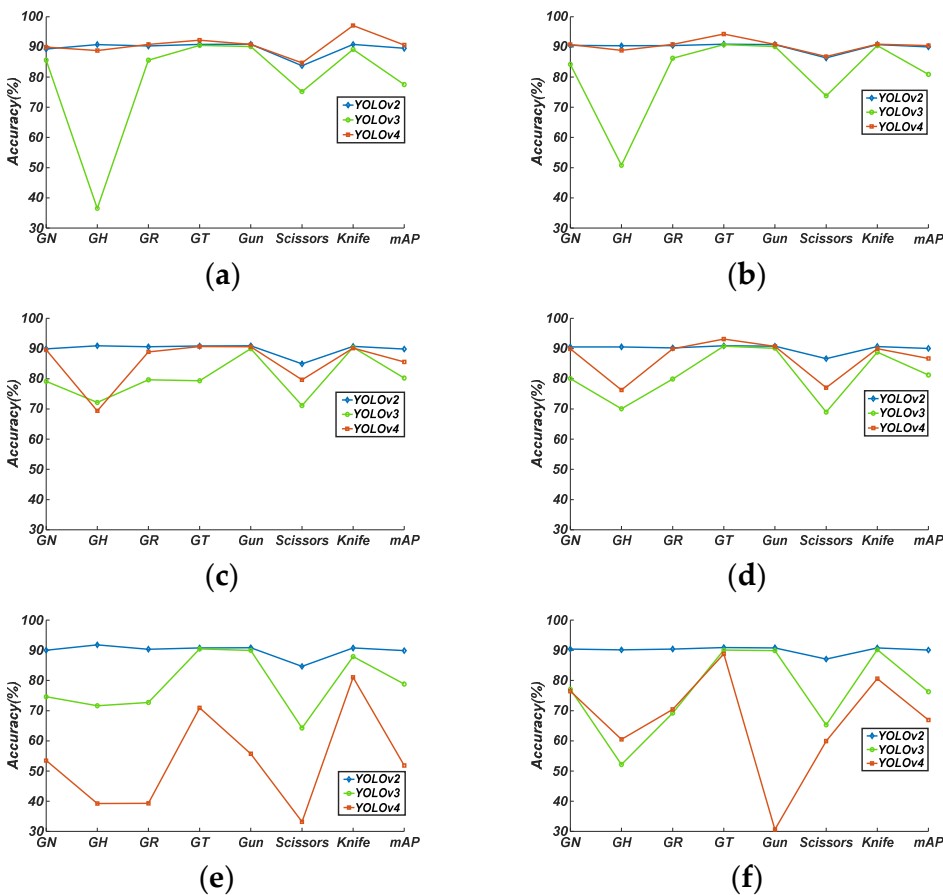

**Figure 11.** Detection accuracy at different learning rates and numbers of iterations: (**a**) learning rate = 0.001, iterations = 9000; (**b**) learning rate = 0.001, iterations = 14,000; (**c**) learning rate = 0.003, iterations = 9000; (**d**) learning rate = 0.003, iterations = 14,000; (**e**) learning rate = 0.005, iterations = 9000; (**f**) learning rate = 0.005, iterations = 14,000.

### 4.2. Comparison of Effects of Various Networks in Dangerous Goods Target Detection

To more intuitively compare the detection effects of the YOLO v2, YOLO v3, and YOLO v4 network models on the seven types of dangerous goods, the model obtained after training was used to test the images in the test set. As described in Section 4.1, we selected the three models corresponding to the maximum mAP to evaluate and compare the actual test results.

The highest mAP values obtained by the YOLO v2, YOLO v3, and YOLO v4 models were 90.07%, 81.26%, and 90.62%, respectively. We compared the test results of six groups. In Figure 12, the red circles mark dangerous goods that were missed. For example, as shown in Figure 12a, YOLO v3 did not detect the scissors and GN, and in Figure 12b, YOLO v3 did not detect the partially obscured GN. Figure 12c shows the side view of Figure 12b. From this perspective, the three explosives (i.e., GH, GR, GN) are highly overlapped, and the three network models did not detect the severely occluded explosive (GR). In addition, the characteristics of scissors are not apparent due to placement position, which also affected model performance; thus, the obtained confidence scores were low. In Figure 12d, the explosive (GR) on the lower right side of the image was not recognized by the YOLO v3 and YOLO v4 models, multiple targets were missed by the YOLO v3 model, and there was a false detection (GH) by the YOLO v4 model. In Figure 12e, the location box of the scissors obtained by the YOLO v2 and YOLO v4 models is inaccurate. Here, the YOLO v3 model

did not detect the gun and scissors, and in Figure 12f, only the gun was detected by the YOLO v3 model, and the YOLO v4 model did not detect one pair of scissors and gun.

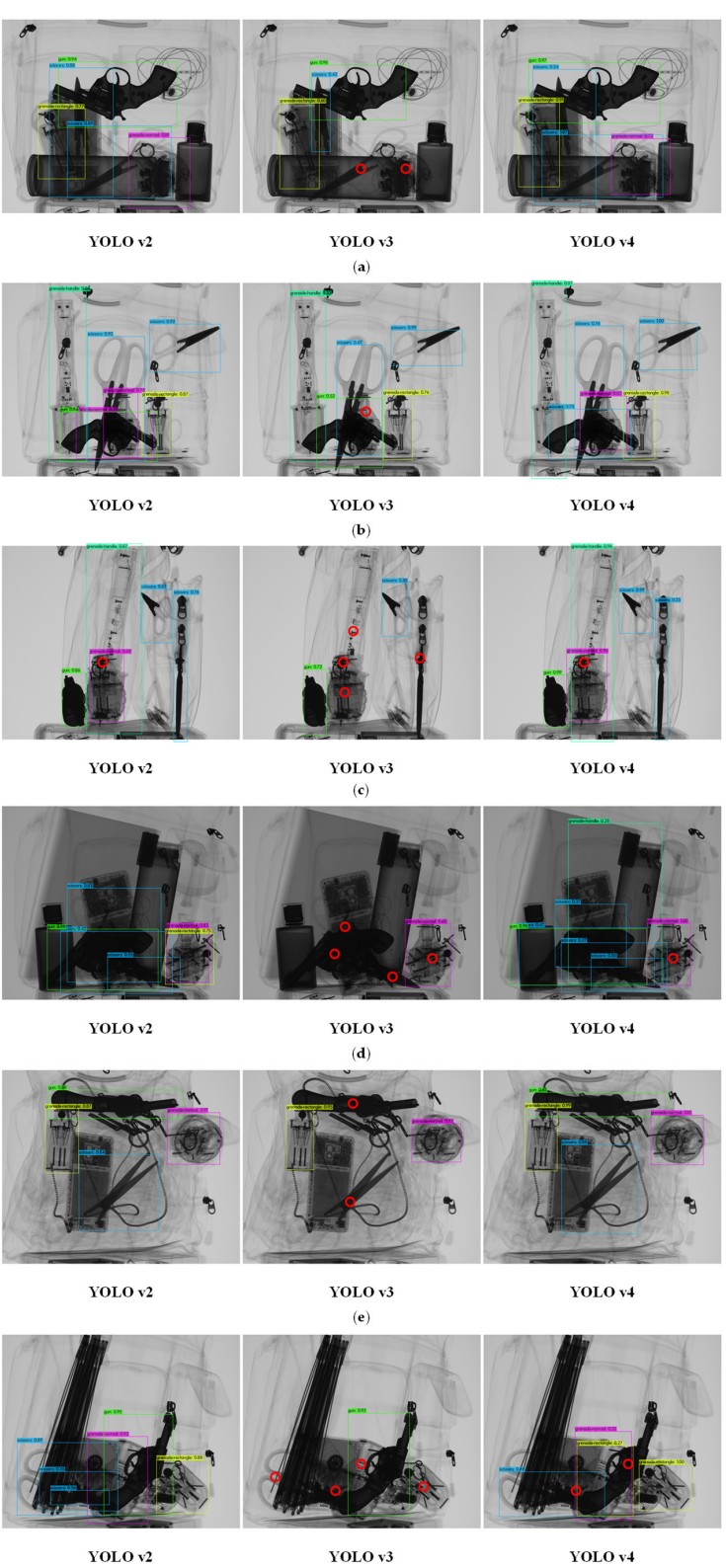

**Figure 12.** Comparison of detection results: (**a**) test group 1; (**b**) test group 2; (**c**) test group 3; (**d**) test group 4; (**e**) test group 5; (**f**) test group 6. (The red circles mark dangerous goods that were missed).

### 4.3. YOLO-T Deep Learning Network Target Recognition

The proposed YOLO-T network was trained on the GDXray-Expanded dataset. The detection accuracies and mAP values for all types of dangerous goods obtained by the proposed YOLO-T network are shown in Table 1. Table 1 also shows the highest detection accuracies of YOLO v2, YOLO v3, and YOLO v4 (Section 4.1). As can be seen, the detection accuracy of the proposed YOLO-T network for a single type of dangerous goods was greater than 95%, which is greatly improved compared with YOLO v3. This proves that using a transformer model as the backbone is better than the DarkNet-53 network in the original YOLO v3 when detecting grayscale images in the X-Ray security inspection task. From the perspective of comprehensive mAP indicators, the proposed YOLO-T network model demonstrates high competitiveness.

**Table 1.** Comparison of accuracy of different deep learning networks.

| Net-Name | GN | GH | GT | GR | Gun | Scissors | Knife | mAP |
|---|---|---|---|---|---|---|---|---|
| YOLO v2 | 0.9014 | 0.9039 | 0.9043 | 0.9091 | 0.9081 | 0.8707 | 0.9075 | 0.9007 |
| YOLO v3 | 0.7007 | 0.8008 | 0.7993 | 0.9083 | 0.9013 | 0.6895 | 0.8881 | 0.8126 |
| YOLO v4 | 0.8876 | 0.8994 | 0.9085 | 0.9221 | 0.9080 | 0.8472 | 0.9708 | 0.9062 |
| YOLO-T | 0.9741 | 0.9633 | 0.9841 | 0.9872 | 0.9880 | 0.9512 | 0.9891 | 0.9773 |

The trained model was used to detect some one-to-one images corresponding to those in Section 4.2. The results are shown in Figure 13. As can be seen, no dangerous goods are not missed, and the location box is very close to the actual position of the objects. In addition, the partially occluded explosive GN in Figure 13a,b was effectively detected. In addition, the severely occluded explosive GR in Figure 13c was well detected. As shown in Figure 13d, the confusion of scissors was improved, and in Figure 13f, the cross-placed scissors were also distinguished accurately.

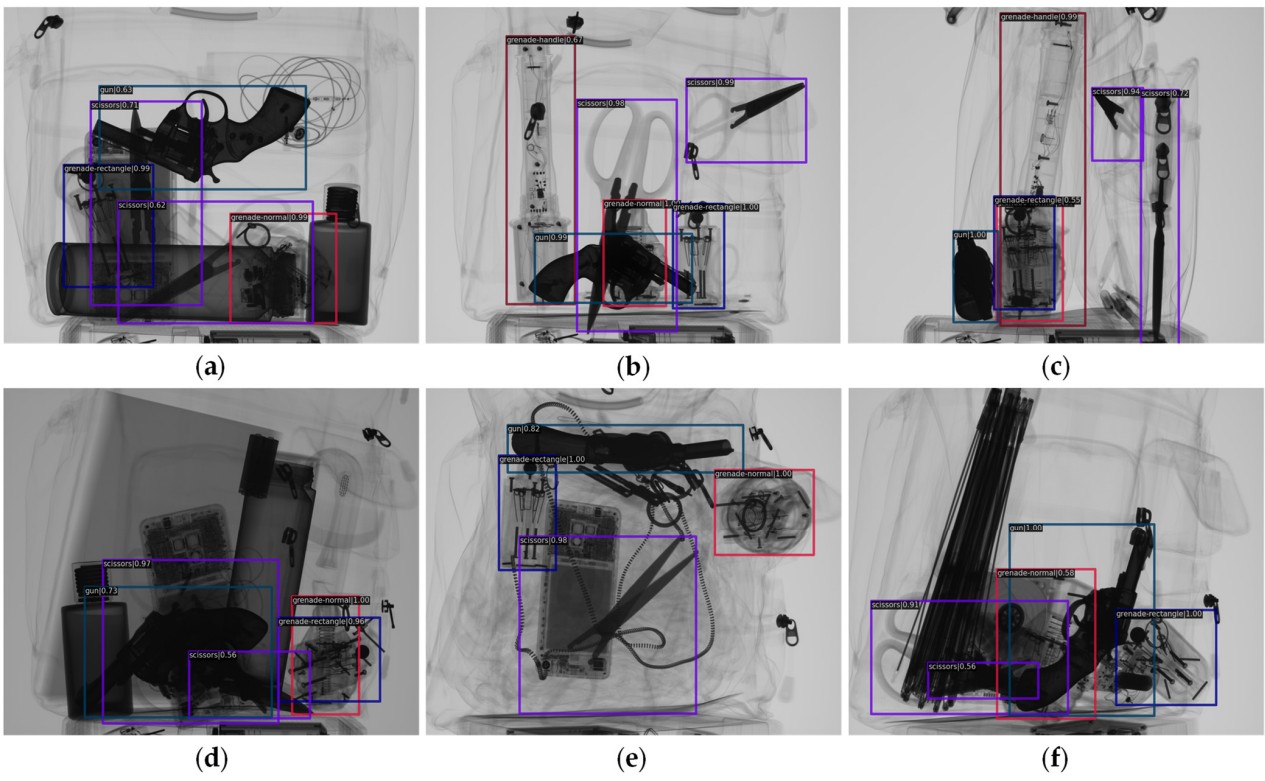

**Figure 13.** Detection results obtained by proposed YOLO-T network model: (**a**) test image 1; (**b**) test image 2; (**c**) test image 3; (**d**) test image 4; (**e**) test image 5; (**f**) test image 6.

Compared to the detection results of the YOLO network models, the detection accuracy of the proposed YOLO-T network was improved significantly relative to the detection of every type of dangerous good. This proves that the proposed YOLO-T network effectively addresses problems related to occlusion and multitarget recognition of X-ray security inspection images. The proposed network realizes accurate identification and target detection of dangerous goods in X-ray images. In addition, the transformer model used as the network backbone employs an attention mechanism rather than a convolution operation to extract features from the images, which effectively improves the detection performance for severely occluded and overlapping targets in X-ray images of dangerous goods.

Here, we do not compare the training time of different types of models, because the training time is generally long and is affected by hardware devices. Compared with the training time, the detection time performance of these models is more important in actual detection. Therefore, we compared the detection time of different models for dangerous goods here. As shown in Figure 14, we tested the detection time of different methods under the same test dataset from Figure 13a–f. Through this experiment, we found that the average time spent by the YOLO v2 method to detect the target of dangerous goods is the shortest, followed by the YOLO-T method, and the YOLO v4 method takes the longest time. According to our previous discussion, the complexity of the YOLO v2 network is the lowest, and the relatively simple network structure saves a lot of time for the detection target. Nevertheless, the YOLO-T method proposed in this paper has the best performance and achieved competitive experimental results in the test of detection time. Considering that the complexity of the YOLO-T method is not lower than the YOLO v3/v4 method, the results show that YOLO-T method has better detection efficiency. In actual engineering applications, the testing time within 30 milliseconds is acceptable. Therefore, the above discussion can show that the YOLO-T method has the advantage of high efficiency in the field of dangerous goods target detection.

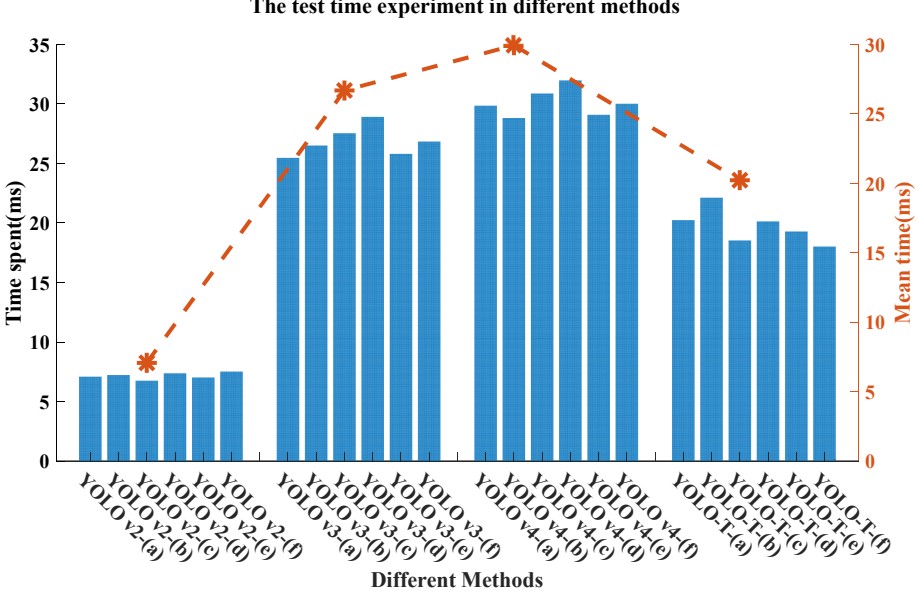

**Figure 14.** Detection time experiment results obtained by different methods.

## 5. Conclusions

In this paper, we proposed the YOLO-T network model to address current limitations in neural network-based X-ray security detection. In addition, we described the GDXray-Expanded dataset, which was constructed specifically for the target task. Through experiments, the following conclusions are drawn in this paper.

(1)　Among the existing YOLO series of deep learning networks, under the background of X-ray security inspection, the deep learning network processing effect of YOLO v2

version is the best, and the optimization of subsequent versions has not significantly improved the detection effect in this research field.

(2) The proposed YOLO-T deep learning network was evaluated experimentally and was compared to existing YOLO network models, which are limited in terms of occlusion recognition and multitarget detection. We found that the proposed YOLO-T network solves these problems by introducing a transformer structure. The YOLO-T deep learning network was able to accurately detect seven different types of dangerous goods.

(3) The YOLO-T deep learning network proposed in this paper can not only detect the seven types of dangerous goods mentioned in GDXray-Expanded. This method can be used to quickly and automatically detect dangerous goods in actual security detection scenarios, which has high engineering application value.

In the future, we plan to study the recognition of dual-energy pseudocolor images and establish a recognition module that can be employed in security inspection equipment.

**Author Contributions:** Conceptualization, M.W. and B.Y.; methodology, M.W.; software, X.W.; validation, M.W., B.Y. and X.W.; formal analysis, C.Y.; investigation, J.X.; resources, B.M.; data curation, K.X.; writing—original draft preparation, M.W.; writing—review and editing, B.Y.; visualization, Y.L.; supervision, C.Y.; project administration, K.X.; funding acquisition, B.Y. All authors have read and agreed to the published version of the manuscript.

**Funding:** National Natural Science Foundation of China (12005157); 2021 Tongji University Excellent Experimental Project (2021-12).

**Institutional Review Board Statement:** Not applicable.

**Informed Consent Statement:** Not applicable.

**Data Availability Statement:** The data used to support the findings of this study are available from the corresponding author upon request.

**Conflicts of Interest:** The authors declare no conflict of interest.

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
