# Peer review of "YOLO-T: Multitarget Intelligent Recognition Method for X-ray Images Based on the YOLO and Transformer Models"

_applsci, doi:10.3390/app122211848_

Round 1
Reviewer 1 Report
This Paper is better defined and the Parameter Setting is good.
Abstract up to mark discussion but Need for minor spell check is required in this paper because of some spelling mistakes as well as grammar errors.
Reviewer 2 Report
Dear Authors,
I request several corrections before a publication is possible:
The Transformers invented by Google Brain has more than 100 Mio hits with Google, but you did not provide any reference to this research. Instead, you just copied figures (your fig. 3) and formulas (your formulas 4 to 9) 1:1 from Vasvani, NIPS 2017, without giving any reference to him. Unfortunately, in your formulas (4) and (5) you have there a 10 000, to what this number is related? Only to your copy source, read there....
I cannot understand why 7 Chinese authors are not able to see this...
Fig. 4 starts with a RGB optical image. You discuss only gray scale images form X-ray, how deep are they? 8 bit ore 16 bit, this is nowhere mentioned.
So please provide the copy source of Fig. 4 or adapt is to your X-ray images.
Fig. 5 Swin Transformer Blocks (STB), I found in Liu, Swin-Transformer 2021, exactly in Fig. 3b, I cannot find a reference in your paper!
So please, make crystal clear what you copied and what you did yourself. This is called scientific respect on the work others....
You combined and expended the GDXray set and the SIXray dataset with your own images, but I was not able to find any info on your final data set you usaed for your investigations. GDXray has about 8000 images containing threats, SIXray more than 1 million. Why it was necessary to complete this vast number if images by own exposures of what? This should be explained somehow.
For your own exposures you used an Xray tube of max.120 kV, but at this voltage NO pair effect happens for electrons. This will need more than 1.024 MeV, you cannot reach with any X-ray tube, but with Isotopes, Betatrons or LINACs. The digital detector you used had an integration time of 70 micro seconds, which I cannot believe. Typically you average several images to improved the SNR and the image exposure time is in the range of 50ms to 500ms. This should be the case in China too, please verify.
Fig. 8s is called "Transmission imaging optical path diagram" but I see there "Set-up for penetration imaging"
in your paper I find several time different "Darknet", but nowhere a reference what this is. Are you aware that this term is also used for the criminal part of the Internet?
You provide in Fig. 14 detection times for threat detection in a single image (it is not given there, but I just assume this...), unfortunately I cannot find any details on the time for training of the networks, which should be considerable larger (maybe days?).
Fig. 12 is interesting, but these are too many images for one page. If you only shown the lower half wit h3 different object combinations, is still demonstrates than Yolo-3 has the poorest performance and Yolo-2 the best.
Finally, the improvement using the YOLO-T network for this type of threat detection and location is really impressive. But nevertheless, a minimum of respect for the work of colleagues by correct citation should be maintained.
Reviewer 3 Report
X-ray security inspection processes have a low degree of automation, long detection times, and are subject to misjudgment due to occlusion. To address these problems, this paper proposes a multi-objective intelligent recognition method for X-ray images based on the YOLO deep learning network and an optimized Transformer structure (YOLO-T)This work need to address some address revisions/concerns before final publication.
1. What is novelty of the work. Please underscore the scientific value added/contributions of your paper in your abstract and introduction and address your debate shortly in the abstract.
2. A good article should include, (1) originality, new perspectives or insights; (2) international interest; and (3) relevance for governance, policy or practical perspective.
3. The work is devoted to an actual scientific and applied problem, performed by correct modern methods and the results are not in doubt. But the presentation and discussion of the results, as well as the conclusions, need to be improved.
4. Literature review section need to be enhanced. Add some other deep learning related works. Some suggested works are :
Diwan, T., Anirudh, G., & Tembhurne, J. V. (2022). Object detection using YOLO: challenges, architectural successors, datasets and applications. Multimedia Tools and Applications, 1-33.
Jiang, P., Ergu, D., Liu, F., Cai, Y., & Ma, B. (2022). A Review of Yolo algorithm developments. Procedia Computer Science, 199, 1066-1073.
Li, Y., Zhang, X., & Shen, Z. (2022). YOLO-Submarine Cable: An Improved YOLO-V3 Network for Object Detection on Submarine Cable Images. Journal of Marine Science and Engineering, 10(8), 1143.
5. Provide architecture of YOLO-CNN.
6. How many epochs are used for training.
7. Discuss optimizers used for training of model.
8. Revise the conclusion section.
9. Any preprocessing/ augmentation was performed over images? If yes, discuss.
